# Recycling and Updating an Educational Robot Manipulator with Open-Hardware-Architecture

**DOI:** 10.3390/s20061694

**Published:** 2020-03-18

**Authors:** Antonio Concha Sánchez, Juan Felipe Figueroa-Rodríguez, Andrés Gerardo Fuentes-Covarrubias, Ricardo Fuentes-Covarrubias, Suresh Kumar Gadi

**Affiliations:** 1Facultad de Ingeniería Mecánica y Eléctrica, Universidad de Colima, Coquimatlán Colima 28400, Mexico; aconcha@ucol.mx (A.C.S.); jfigueroa_@ucol.mx (J.F.F.-R.); fuentesg@ucol.mx (A.G.F.-C.); fuentesr@ucol.mx (R.F.-C.); 2Facultad de Ingeniería Mecánica y Eléctrica, Universidad Autónoma de Coahuila, Torreón, Coahuila 27276, Mexico

**Keywords:** open-source hardware, educational robot, artificial vision, decentralized control, recycled robot, robot control

## Abstract

This article presents a methodology to recycle and upgrade a 4-DOF educational robot manipulator with a gripper. The robot is upgraded by providing it an artificial vision that allows obtaining the position and shape of objects collected by it. A low-cost and open-source hardware solution is also proposed to achieve motion control of the robot through a decentralized control scheme. The robot joints are actuated through five direct current motors coupled to optical encoders. Each encoder signal is fed to a proportional integral derivative controller with anti-windup that employs the motor velocity provided by a state observer. The motion controller works with only two open-architecture Arduino Mega boards, which carry out data acquisition of the optical encoder signals. MATLAB-Simulink is used to implement the controller as well as a friendly graphical interface, which allows the user to interact with the manipulator. The communication between the Arduino boards and MATLAB-Simulink is performed in real-time utilizing the Arduino IO Toolbox. Through the proposed controller, the robot follows a trajectory to collect a desired object, avoiding its collision with other objects. This fact is verified through a set of experiments presented in the paper.

## 1. Introduction

Robot manipulators are one of the most widely used mechatronic systems in the industry, whose applications include the assembly of elements, as well as the welding and painting of parts. Due to its great usefulness in the industry, it is very important to study its kinematics, dynamics, and automatic control in engineering careers related to mechatronics and robotics. A characteristic of robot manipulators is that they are usually manufactured with a closed architecture in their automatic control. Once a robot meets its end-of-Life, it is resold, reused, or recycled, which are known as the “3Rs” [1]. A manipulator is usually classified as unusable equipment when its controller is damaged. The reason is that the cost of its reparation can be expensive. In this case, it could be convenient to propose a low cost methodology to re-manufacture the robot, where its mechanical components can be reused and its control system is redesigned using an open architecture.

In the literature, there are several motion controllers for robot manipulators, some of which are recycled and are employed in experimental educational platforms to validate the theory seen in class. Bomfim et al. [2] re-manufactured the controller of a robot manipulator for the automotive industry, whose trajectories are designed with the MATLAB and Mach3 programs. Sanfilippo et al. [3] and Soriano et al. [4] recycled robotic arms for automation engineering education. The robot in [3] is useful for student academic training, whose controller cabinet was developed by students using a PLC architecture. On the other hand, the robot in [4] is built with recycled LEGO pieces, and it is controlled with an Arduino Mega board, which is programmed using the Simulink Support Package for Arduino Hardware. Yen et al. [5] developed a low-cost collaborative robot that employs a virtual force sensor and stiffness control to safety collision detection and low-precision force control. The authors of [6,7,8] presented educational robot manipulators, whose movements are carried out by means of radio control servomotors that have controllers that cannot be modified. The manipulator in [6] is operated from a graphical interface, while the robotic arm in [7] has two cameras to detect, collect, and move objects. On the other hand, Cocota et al. [8] described the design and development of a 4-DOF manipulator with a low cost of approximately USD 150. Robot manipulators based on Dynamixel servomotors are developed in [9,10], where Rivas et al. [9] presented the control system of a 6-DOF manipulator controlled through the Python software, whereas Kim and Song [10] designed a mechanism to counterbalance the gravitational torques of a 5-DOF robot arm. Manzoor et al. [11] developed an experimental platform called AUTAREP, which consists of a robotic arm, model ED7220C, from the ED Corporation. The authors of [11] replaced the original controller of the manipulator, which has a closed architecture, with a controller manufactured by them. This controller is described in [12], has an open architecture, is programmed through a graphical user interface (GUI), and it has been implemented as: PID regulator [13], computed torque controller [14], and optimal regulator [15]. On the other hand, sliding mode controllers, an adaptive regulator, and neural networks are, respectively, proposed in [16,17,18,19,20,21], for tracking control of robot manipulators employed for educational and research purposes. In the literature, the development of virtual or simulated robot manipulators is also proposed; for example, the authors of [22,23,24,25] presented robust controllers validated in simulations with the so-called PUMA 560 robot manipulator. However, these manipulators usually do not contemplate friction and backlash, which are present in a real manipulator and cause tracking errors, limit cycles, and other problems that directly affect the manipulator’s motion control.

This article presents an experimental educational platform based on a recycled 4-DOF robotic arm with gripper, which is employed to teach and study its kinematics, dynamics, and automatic control. The recycled robot reuses the mechanical parts and motors of a manipulator from ED Corporation, model ED7220C, whose controller was damaged. Since its repair cost is high, a in-house design is considered. The proposed experimental platform is an upgraded version of our first work described in [26] to which several capabilities has been added such as: a force sensor inside the robot gripper to detect objects; artificial vision to locate objects and to pick them up according to its shape and color; an anti-windup technique to a PID controller to improve transient response of the movements; a fine tuning of the controller gains to reduce tracking errors; a graphical interface to interact with a user, and trajectory planning to avoid the collision of the robot with objects. All the programming of the recycled robot is carried-out in MATLAB Simulink. Its motion controller has a decentralized scheme that does not take into account the robot dynamics, and it is applied to five direct current (dc) motors coupled to the robot joints, whose positions are detected by the optical encoders. A parameter identification methodology based on the Recursive Least Squares method is also designed to estimate the parameters of the dc motors, which are subsequently employed to design their controllers and state observers that estimate the joint velocities. Data acquisition of the encoders is realized by two Arduino Mega boards. The communication between these boards and MATLAB-Simulink is carried out in real-time using the open-source ARDUINO IO Toolbox [27]. The proposed controller has the advantage that it is programmed with a visual environment based on block diagrams that has a higher level of abstraction than the programming language used in the AUTAREP platform [11]. Furthermore, in comparison with the on by Soriano et al. [4], the proposed motion controller is developed with a Simulink toolbox that reads encoder signals, thus simplifying their acquisition. Programming the controller and the artificial vision in Simulink has the advantage of monitoring all signals of the controller by means of scopes, and of using blocks that facilitate the design of other control algorithms such as robust, optimal, adaptable, fuzzy, and neural networks, among others. It is worth mentioning that the proposed experimental educational platform is a key element of the Robotics Laboratory of the Faculty of Mechanical and Electrical Engineering (FIME) at the Universidad de Colima in Mexico, where undergraduate students validate the theory seen in courses of robotics and automatic control, and they also use the robot for research purposes. For example, it was used by three undergraduate students during their final degree projects, whose achievements are reflected in this manuscript. Similarly, the robot has also been used in internal workshops to motivate students to join and remain at the FIME, as well as to show them the importance of robotics and automatic control.

The article is organized as follows. Section 2 describes the architecture of the recycled robot manipulator. Its kinematics and dynamics are presented in Section 3 and Section 4, respectively. Section 5 shows the parameter identification of the robot actuators, and their parameter estimates are used in Section 6 for the design of PID controllers and state observers. On the other hand, the trajectory planning, the artificial vision, the GUI interface, and experimental experiments are discussed in Section 7, Section 8, Section 9 and Section 10, respectively. Finally, Section 11 establishes the conclusions of the manuscript.

## 2. Robot Architecture

The achitecture of the recycled robot manipulator is shown in Figure 1. It consists of a 4-DOF robotic arm, model ED7220C, developed by the ED Corporation from Korea. Its joints are shown in Figure 2, which are located at the body, shoulder, elbow, and wrist. The manipulator also has a gripper to collect objects, and, over it, there is a resistive force sensor, model FSR 402 from the Interlink Electronics company of USA. This sensor determines if an object is inside the gripper. All joints and gripper have limit switches, which indicate their minimum and maximum displacements. Moreover, these switches allow establishing the manipulator initial position. To achieve this position, the manipulator also has an 11.43 cm flex sensor from the American company SparkFun Electronics; this sensor is located at the elbow joint. The body, shoulder, and elbow joints are coupled to permanent dc motors, model DME38B50G-116 from the company Servo of Japan. In the sequel, these joints are denoted as $q_{1}$, $q_{2}$, and $q_{3}$, respectively. On the other hand, the wrist and gripper are, respectively, driven by DME38B50-115 and DME33B37G-171 dc motors also from Servo. The wrist joint, denoted as $q_{4}$, is actuated by a differential gear mechanism [28] coupled to two dc motors. Each motor of the robot includes an optical encoder to determine its position and is connected to a gearbox to increase its torque while reducing its speed.

Two Arduino Mega boards, from the Italian company Arduino, control the position of the motors. Each board acquires the position data of three motors, and it is communicated with a personal computer through a USB connection. The control signal of each motor is produced by the program MATLAB-Simulink from the American corporation Mathworks. This signal is communicated to the Arduino Mega board via the Arduino IO toolbox, which converts the control signal to Pulse Width Modulation (PWM). The PWM signals are fed to L298N Dual H-Bridge Driver Modules from the Chinese corporation Haitronic. These modules provide the power to the dc motors. On the other hand, a webcam from the Swiss company Logitech, model C525, provides artificial vision to the manipulator and permits obtaining the positions of the objects taken by the manipulator. Image processing is carried out in MATLAB-Simulink using the Computer Vision Toolbox. A user-friendly interface GUI, created in MATLAB, permits selecting the shape and color of the objects taken by the manipulator, whose kinematics is described below. It is worth mentioning that the main components of the proposed controller, such as webcam, Arduino boards, motor drivers, force and flex sensors, connectors, and cables, have a total cost of about 230 USD.

## 3. Robot Kinematics

Figure 3 shows the structure of the manipulator, as well as its 4-DOF $q_{1},q_{2},q_{3},q_{4}$, which are the joint positions. This figure also shows the coordinates (x,y,z) of the end-effector, as well as the angle ϕ that specifies its orientation.

### 3.1. Forward Kinematics

Using the trigonometric relationships between the links and their lengths leads to the forward kinematics of the manipulator, which provides the position of the end-effector with respect to the joint positions. It is given by:(1)$$\begin{array}{cc} x & =C_{1}\left(l_{1}\;\cdot \;C_{2}+l_{2}\;\cdot \;C_{23}+l_{3}\;\cdot \;C_{234}\right)\\ y & =S_{1}\left(l_{1}\;\cdot \;C_{2}+l_{2}\;\cdot \;C_{23}+l_{3}\;\cdot \;C_{234}\right)\\ z & =d_{1}+l_{1}\;\cdot \;S_{2}+l_{2}\;\cdot \;S_{23}+l_{3}\;\cdot \;S_{234} \end{array}$$
where $d_{1}=370$ mm, $l_{1}=l_{2}=220$ mm, and $l_{3}=140$ mm are the lengths of the links. Furthermore, we used the following shorthand notations
$$C_{i}=\cos\left(q_{i}\right);\;S_{i}=\sin\left(q_{i}\right);\;C_{iwj}=\cos\left(q_{i}+q_{w}+q_{j}\right);\;S_{iwj}=\sin\left(q_{i}+q_{w}+q_{j}\right)$$

### 3.2. Robot Workspace

The workspace is the volume that can reach the end-effector, and it is constructed through the forward kinematics in Equation (Equation 1), as well as the range covered by the joints of the manipulator, as shown in Table 1.

Figure 4 shows the arm movement range in the plane xy at a height of 115 mm with respect to the robot base. The figure shows that this range encompasses a radius of approximately 400 mm. The objects manipulated by the robot are placed over a rectangle area *A* located in front of the robot base. The dimensions *n* and *l* of this area are determined analytically to maximize it, as follows. Note that *n* and *l* are given by:(2)l=rsin(α),p+n=rcos(α)
where r=400 mm, p=200 mm, and α is an unknown angle to be determined. Therefore, the area *A* can be written as
(3)A=[rcosα−p][rsinα]

The derivative of Equation (Equation 3) with respect to α is given by
(4)$$\frac{dA}{d\alpha }=r\left(r[\cos^{2}\alpha -\sin^{2}\alpha ]-p\cos\alpha \right)=r\left(r[2\cos^{2}\alpha -1]-p\cos\alpha \right)=r\left(2r\cos^{2}\alpha -p\cos\alpha -r\right)$$

By equaling this derivative to zero produces the critical point α=0.568 rad = 32.5°, which gives the maximum rectangular area *A*. For simplicity, a value of α=0.568 rad = 30° is used that yields the dimensions *n* = 200 mm, *l* = 140 mm, and the area *A* shown in Figure 5, which is represented as a purple rectangle. Note that this figure also shows that the robot camera is located 585 mm above the area *A*.

### 3.3. Inverse Kinematics

The section describes the inverse kinematics of the manipulator whose objective is to obtain the joint positions $q_{1}$, i=1,…,4 so that the end-effector is placed in a specific position and orientation. The inverse kinematics of the robot is given by [29]:(5)$$\begin{array}{cc} d_{2} & =z-d_{1}\\ \rho & =\sqrt{x^{2}+y^{2}}\\ p_{wx} & =\rho -l_{3}\cos\left(\varphi \right)\\ p_{wy} & =d_{2}-l_{3}\sin\left(\varphi \right)\\ D & =\frac{p_{wx}^{2}+p_{wy}^{2}-l_{1}^{2}-l_{2}^{2}}{2l_{1}l_{2}}\\ q_{1} & =\mathrm{atan}2(y,x)\\ q_{2}=\mathrm{atan}2\left(p_{wy},p_{wx}\right)- & \mathrm{atan}2\left(l_{2}\sin\left(q_{3}\right),l_{1}+l_{2}\cos\left(q_{3}\right)\right)\\ q_{3} & =\mathrm{atan}2\left(\sqrt{1-D^{2}},D\right)\\ q_{4} & =\phi -q_{2}-q_{3} \end{array}$$
where $\mathrm{atan}2(y^{*},x^{*})$ represents the arctangent of $y^{*}/x^{*}$ and takes into account the sign of each argument to determine the quadrant corresponding to the angle between $x^{*}$ and $y^{*}$.

The next section presents the dynamic equation of the manipulator that takes into account the torques required for the execution of the robot motion.

## 4. Robot Dynamics

The dynamic behavior of the manipulator is described by the following expression [29]
(6)$$M\left(q\right)\ddot{q}+C\left(q,\dot{q}\right)\dot{q}+g\left(q\right)+f\left(\dot{q}\right)=\tau$$
where $q={\left[q_{1},q_{2},q_{3},q_{4}\right]}^{T}$ is the vector of joint positions; $\dot{q}$ is the angular velocity vector, $M\left(q\right)=M{\left(q\right)}^{⊺}\in R^{4\times 4}$ is called inertia matrix, $C\left(q,\dot{q}\right)\dot{q}\in R^{4\times 1}$ represents a centrifugal and Coriolis force vector, and $g\left(q\right)\in R^{4\times 1}$ is a gravitational forces vector. In addition, $f\left(\dot{q}\right)\in R^{4\times 1}$ is a frictional forces vector and $\tau \in R^{4\times 1}$ is a vector of torques applied by the actuators at the joints.

### 4.1. Dynamic Model of the Actuators

The set of the joint actuators can be represented by the following matrix differential equation [30]:(7)$$\ddot{q}+G\dot{q}+R\tau =\mathrm{Ku}$$
where
(8)$$\begin{array}{cc} G & =\mathrm{diag}\left(a_{1},a_{2},a_{3},a_{4}\right)\\ R & =\mathrm{diag}\left(\frac{1}{J_{m1}r_{1}^{2}},\frac{1}{J_{m2}r_{2}^{2}},\frac{1}{J_{m3}r_{3}^{2}},\frac{1}{J_{m4}r_{4}^{2}}\right)\\ K & =\mathrm{diag}\left(b_{1},b_{2},b_{3},b_{4}\right)\\ u & =\mathrm{diag}\left(u_{1},u_{2},u_{3},u_{4}\right)\\ \tau & =\mathrm{diag}\left(\tau_{1},\tau_{2},\tau_{3},\tau_{4}\right) \end{array}$$

$a_{i}$ and $b_{i}$, i=1,2,3,4 are positive parameters. $u_{i}$, $\tau_{i}$, $J_{mi}$, and $r_{i}$ are the input voltage, load torque, motor inertia, and gear reduction ratio of the *i*th joint actuator, respectively. diag(p) represents a diagonal square matrix with the elements of *p* in the main diagonal.

### 4.2. Mathematical Model of the Robot Manipulator with Actuators

Substituting τ of Equation (Equation 6) into Equation (Equation 7) yields
(9)$$\left[RM\left(q\right)+I\right]\ddot{q}+RC\left(q,\dot{q}\right)\dot{q}+Rg\left(q\right)+Rf\left(\dot{q}\right)+G\dot{q}=Ku$$
where I is the identity matrix of size 4×4. The previous model is considerably reduced when the gear ratios $r_{i}$, i=1,2,3,4 are high, i.e., $r_{i}\gg 0$. In this case, R≈O and Equation (Equation 9) approximates to:(10)$$\ddot{q}+G\dot{q}=\mathrm{Ku}$$

The gear ratios *r* of the dc motors corresponding to the body, shoulder, elbow, and wrist of the manipulator are 720, 576, 576, and 133, respectively. Since the gear ratios of these motors are high, the dynamics of the manipulator in Equation (Equation 6) can be neglected. Therefore, an independent controller can be designed for each robot joint using the linear model in Equation (Equation 10).

Parameters $a_{i}$ and $b_{i}$ of the dc motor models are unknown, and they are estimated using the recursive least squares algorithm described in the following section.

## 5. Parameter Identification of the Actuators

The Recursive Least Squares algorithm (RLSM) [31] is used to identify the parameters of the robot actuators, which permit designing: (1) controllers to obtain high precision movements in the manipulator; and (2) state observers to estimate the motor speed. Moreover, the parameter identification is necessary to simulate the robot manipulator and to detect faults on it [32]. To estimate the parameters $a_{i}$ and $b_{i}$, the actuators are operated in closed loop using a proportional controller and a sinusoidal reference input signal.

Since signals ${\dot{q}}_{i}\left(t\right)$ y ${\ddot{q}}_{i}\left(t\right)$ of the model in Equation (Equation 10) are not available, parameters $a_{i}$ and $b_{i}$ are estimated using only measurements of the motor voltage $u_{i}$ and its position $q_{i}$. To this end, each uncoupled model in Equation (Equation 7) is filtered by means of the filter $H\left(s\right)=\lambda_{2}/\left(s^{2}+\lambda_{1}s+\lambda_{2}\right)$, $\lambda_{1},\lambda_{2}>0$, which also attenuates measurement noise, thus minimizing its effect in the parameter identification algorithm. This filtering procedure produces:(11)$$z_{i}\left(t\right)=\psi_{i}^{⊺}\left(t\right)\theta_{i}$$
where
(12)$$\begin{array}{cc} z_{i}\left(t\right) & =L^{-1}\left[s^{2}H\left(s\right)Q_{i}\left(s\right)\right]\\ \psi_{i}\left(t\right) & =\left[\begin{array}{c} L^{-1}\left[-sH\left(s\right)Q_{i}\left(s\right)\right]\\ L^{-1}\left[H\left(s\right)U_{i}\left(s\right)\right] \end{array}\right]\\ \theta_{i} & =\left[\begin{array}{c} a_{i}\\ b_{i} \end{array}\right] \end{array}$$

L and $L^{-1}$ are the Laplace operator and its inverse, respectively; similarly, $Q_{i}\left(s\right)=L\left[q_{i}\left(t\right)\right]$ and $U_{i}\left(s\right)=L\left[u_{i}\left(t\right)\right]$.

Signals $z_{i}\left(t\right)$ and $\psi_{i}\left(t\right)$ in Equation (Equation 12) are sampled every $T_{s}$ seconds, and they are used by the RLSM given by [31]:(13)$$\begin{array}{cc} {\hat{\theta }}_{i}\left(k\right) & ={\hat{\theta }}_{i}\left(k-1\right)+P_{i}\left(k\right)\psi_{i}\left(k\right)\epsilon_{i}\left(k\right)\\ P_{i}\left(k\right) & =\frac{1}{\gamma_{i}}\left[P_{i}\left(k-1\right)-\frac{P_{i}\left(k-1\right)\psi_{i}\left(k\right)\psi_{i}^{T}\left(k\right)P_{i}\left(k-1\right)}{\gamma_{i}+\psi_{i}^{T}\left(k\right)P_{i}\left(k-1\right)\psi_{i}\left(k\right)}\right]\\ \epsilon_{i}\left(k\right) & =z_{i}\left(k\right)-\psi_{i}^{T}\left(k\right){\hat{\theta }}_{i}\left(k-1\right) \end{array}$$
where ${\hat{\theta }}_{i}\left(k\right)={\left[{\hat{a}}_{i}\left(k\right),{\hat{b}}_{i}\left(k\right)\right]}^{T}$ is an estimate of $\theta_{i}$. $\gamma_{i}$ is called forgetting factor and satisfies $0<\gamma_{i}\leq 1$. In addition, variable $P_{i}\left(k\right)=P_{i}^{⊺}\left(k\right)$ is called covariance matrix.

To experimentally identify the parameters $a_{i}$ and $b_{i}$, i=1,2,3,4, the RLSM was configured with the following values: $T_{s}=0.02$ s, $\gamma_{i}=0.997$, $P_{i}\left(0\right)=1000\mathrm{diag}\left[1,1,1,1\right]$, $\lambda_{1}=20$, $\lambda_{2}=100$. Figure 6 shows the time evolution of estimates ${\hat{a}}_{1}$ and ${\hat{b}}_{1}$ corresponding to the base actuator. It is shown that the estimates converge in approximately 1 s. Table 2 shows the estimated parameters of each actuator and its corresponding joint.

## 6. Robot Control

A PID controller is used to regulate the position of the actuators. This controller is a modification of the basic Proportional Derivative Controller (PID), and it is employed to avoid the set-point kick phenomenon, which consists of abrupt changes of the control signal due to sudden changes of the reference input [33]. The PID controller is given by [33]:(14)$$u_{i}\left(t\right)=K_{Pi}e_{i}\left(t\right)+K_{Ii}\int e_{i}\left(\tau \right)d\tau -K_{Di}\frac{dq_{i}\left(t\right)}{dt}$$

Note that the derivative action is applied only to the output signal $q_{i}\left(t\right)$. $e_{i}\left(t\right)$ is the position error of the *i*th joint that is defined as $e_{i}\left(t\right)=q_{di}-q_{i}$, where $q_{di}$ is the desired position of the *i*th joint. Moreover, $k_{Pi}$, $k_{Ii}$, and $k_{Di}$ are, respectively, the proportional, integral, and derivative gains of the *i*th position controller. The Routh–Hurwitz stability criterion [34] allows determining the following range of gains $k_{Pi}$, $k_{Ii}$, and $k_{Di}$ that guarantee a stable closed-loop system.
(15)$$K_{Di}>\frac{-a_{i}}{b_{i}};\;K_{Ii}\geq 0;\;K_{Pi}>\frac{K_{Ii}}{a_{i}+K_{Di}b_{i}}$$

Since the nominal values $a_{i}$ and $b_{i}$ of each motor are not available, the estimates ${\hat{a}}_{i}$ and ${\hat{b}}_{i}$ produced by the RLSM are replaced in Equation (Equation 15).

In order for the integral term of the PID controller not to cause a slow transient position response due to the voltage saturation of the actuators, this term is implemented using the anti-windup compensation scheme in Figure 7, where $K_{ai}$ is the anti-windup gain. In this figure, $u_{\min}$ and $u_{\max}$ denote the minimum and maximum voltage of the actuators, respectively.

The PID controller with anti-windup requires the velocities of the robot actuators, which are not available. However, these signals are estimated by means of a state observer described below.

### State Observer

The model of an actuator in Equation (Equation 10) can be written as the following state space equation:(16)$$\begin{array}{cc} \dot{x} & =Ax+Bu\\ y & =Cx \end{array}$$
where
(17)$$x=\left[\begin{array}{c} x_{1}\\ x_{2} \end{array}\right]=\left[\begin{array}{c} q_{i}\\ {\dot{q}}_{i} \end{array}\right]\;A=\left[\begin{array}{cc} 0 & 1\\ 0 & -a_{i} \end{array}\right],\;B=\left[\begin{array}{c} 0\\ b_{i} \end{array}\right],\;C=\left[\begin{array}{cc} 1 & 0 \end{array}\right]$$

For estimating the speed ${\dot{q}}_{i}$ of the *i*th motor, a Luenberger observer is programmed, whose mathematical model is given by [35]:(18)$$\dot{\hat{x}}=A\hat{x}+Bu+K_{o}\left(y-C\hat{x}\right)=\left(A-K_{o}C\right)\hat{x}+Bu+K_{o}y$$
where $\hat{x}={\left[{\hat{x}}_{1},\;{\hat{x}}_{2}\right]}^{T}$ is an estimate of x, $K_{o}={\left[K_{1},\;K_{2}\right]}^{T}$ is the observer gain, and matrices A and B are constituted with the parameter estimates ${\hat{a}}_{i}$ and ${\hat{b}}_{i}$, respectively. Note that ${\hat{x}}_{2}={\hat{\dot{q}}}_{i}$ is the estimated velocity employed by the PID controller.

Table 3 presents the gains of the PID controllers with anti-windup, as well as the gains of the state observer corresponding to each motor. On the other hand, Figure 8 shows the coupling of the state observer with the PID controller, whose gains $K_{Pi}$, $K_{Ii}$, $K_{Di}$, and $K_{ai}$ are tuned so that the joint response $q_{i}$ under a step input is sufficiently fast and damped. Moreover, the integral action of the actuator controllers corresponding to the shoulder and elbow joints allow counteracting the gravity forces of the links connected to these joints. Likewise, the observer gains are selected to produce an observer dynamics with both poles equal to −6.

The next section describes the trajectory of the end-effector to reach, take, and release an object in the robot workspace. This planned trajectory generates the reference inputs $q_{di}$ to the PID controllers of the actuators, which assure that the robot executes the desired motion.

## 7. Trajectory Planning

Figure 9 shows a flowchart representing the trajectory planning in the Cartesian workspace of the manipulator. The path planning algorithm consist of a sequence of points along the path, which are denoted as A–D. Point A is the robot initial position, Point B is a position above the object, Point C is the object position, andPoint D is the position where the manipulator deposits the object, as illustrated in Figure 10. Through the sequence of points A-B-C-B-A-D, the robot collects an object and deposits it in a container, avoiding collisions with other objects. To execute the trajectory planning, it is necessary to resort to the robot inverse kinematics in order to convert the Cartesian Points A–D into joint input references $q_{di}$ provided to the PID controllers of the actuators. Due to the high gear reduction ratio of the dc motors, the movement from one point to another point is smooth.

To detect the position and shape of the objects gripped by the robot, it has artificial vision, as mentioned below.

## 8. Artificial Vision

The robot has artificial vision through a webcam located at a height of 700 mm above the robot base. The Image Acquisition Toolbox of Simulink is used to provide the artificial vision to the robot, and students can use this powerful tool for image processing, image segmentation [36], image enhancement, visual perception [37], recognition of 3D objects [36], human-like visual-attention-based artificial vision [38], visual SLAM [39], feature extraction [40], and noise reduction, just to mention a few. Camera images are acquired using the block From Video Device (FVD) of this toolbox, which extracts their RGB values in a matrix that can be processed by the armory of matrix operators and functions of MATLAB [41]. Images are acquired at five frames per second (FPS). The image of a red circle on the robot workspace is shown in Figure 11.

### 8.1. Color Detection

The FVD block is configured to only visualize objects within the purple rectangle of Figure 5. For this purpose, a resolution of 419 × 147 pixels is used, where each pixel is equal to 0.955 mm. RGB values obtained from an image are processed to produce a grayscale image for each RGB plane. The gray scale of the red circle in Figure 11 is shown in Figure 12 for each RGB plane.

The grayscale image of each RGB plane is filtered in order to smooth the edges of the objects using the block Median Filter of Simulink. Subsequently, a thresholding is applied to the filtered images so that the manipulator can recognize red and yellow objects. The thresholding produces a binary image, where a value of 1 means white, whereas a value of 0 means black. The thresholding employed to detect red and yellow colors is represented by means of the following expression
(19)$$\begin{array}{c} f_{th}\left(p^{r},p^{g},p^{b}\right)=\left\{\begin{array}{cc} 1 & \mathrm{if}\;p_{\sup\;}^{r}\leq p^{r}\leq p_{\inf\;}^{r}\;\mathrm{and}\;p_{\sup\;}^{g}\leq p^{g}\leq p_{\inf\;}^{g}\;\mathrm{and}\;\mathrm{if}\;p_{\sup\;}^{b}\leq p^{b}\leq p_{\inf\;}^{b}\;\\ 0 & \mathrm{otherwise} \end{array}\right. \end{array}$$
where $f_{th}\left(p^{r},p^{g},p^{b}\right)$ is the output of the thresholding and $p^{j}$, j=r,g,b, represents a pixel in the planes red, green, and blue, respectively. Their upper and lower limits are written, respectively, as $p_{\sup}^{j}$ and $p_{\inf}^{j}$, whose values in Table 4 permit detecting red and yellow colors.

Finally, a morphological operation, executed with the Erosion and Dilation Simulink blocks, permits smoothing the edges of the resulting binary image, thus producing Figure 13.

### 8.2. Shape Detection

To determine the object shape, its area *A*, perimeter *P*, compaction *C*, and centroid $C_{e}$ are calculated. The previous operations, except for compaction, are performed by the block Blob Analysis of Simulink. The object compaction *C* is defined as [42,43,44]:(20)$$C=\frac{P^{2}}{A}$$

The manipulator collects circle and square objects whose compaction is given by:(21)$$C_{\mathrm{square}}=\frac{{\left(4L\right)}^{2}}{L^{2}}=16,\;\;C_{\mathrm{circle}}=\frac{{\left(2\pi R\right)}^{2}}{\pi R^{2}}=4\pi$$
where *L* is the length of the sides of the square and R is the radius of the circle.

Figure 14 briefly describes the artificial vision process and its interaction with the robot motion control.

### 8.3. Correction of the Object Position

The top surface of the objects, which is seen by the camera, has a height with respect to the xy plane where the objects are placed, as shown in Figure 15. For this reason, the position of an object, denoted as $x_{obj}$, does not coincide with the one provided by the camera, defined as $x_{cam}$. The following equation is used to compute $x_{obj}$:(22)$$x_{\mathrm{obj}}=x_{\mathrm{cam}\;}-\frac{h_{\mathrm{obj}\;}}{h_{\mathrm{cam}}}x_{\mathrm{cam}}$$
where $h_{obj}$ and $h_{cam}$ are, respectively, the heights of the object and camera with respect to the robot base. It is important to mention that a similar correction of the object position is carried out on the *y* axis.

## 9. Graphical User Interface

A graphic user interface (GUI) is designed to facilitate the interaction between the user and the recycled manipulator. The GUIDE tool, included in MATLAB to develop high-level graphical and simply layout, was used to design the GUI that is shown in Figure 16. It has buttons to place the manipulator in its initial position, to stop it for emergency, and to select the color and shape of the objects taken by the robot. Moreover, the GUI permits visualizing the positions of the objects in the robot workspace as well as capturing camera images.

## 10. Experimental Results

Experimental results obtained with the proposed controller and artificial vision of the robot manipulator are presented in this section. All experiments used a sampling period $T_{s}$ of 0.06 s. The manipulator’s aim was to collect four cylinders randomly arranged, on that bases of which geometric figures with a shape of circle or square that were yellow or red were attached, as shown in Figure 17. These cylinders weighed 46 g; however, the actuators of the manipulator have enough torque to move payloads up to 1 kg [45]. The MATLAB and Simulink files of the robot control programming were uploaded to the open-source website Github [46], including the instructions to run the proposed controller. Moreover, this website contains videos of the experiments that are shown in this section.

The first experiment consisted of gripping a yellow circle, whose coordinates were x=46 mm, y=297 mm, and z=145 mm, which were obtained with the help of the artificial vision of the robot. The trajectories of the end-effector in the *x*, *y*, and *z* axes, as well as its angle ϕ, are shown in Figure 18. In this figure, $x_{d}$, $y_{d}$, $z_{d}$, and $\phi_{d}$ represent the desired trajectories of the end-effector, where $\phi_{d}$ is fixed to −90°. The yellow circle was gripped by the robot at approximately 14 s, and it was released in a container at around 24 s. This fact is corroborated with the help of Figure 19 that shows the voltage of the force sensor placed inside the robot gripper. A voltage above 0.4 V of this sensor indicates that the object is gripped by the end-effector.

The joint trajectories $q_{1}$, $q_{2}$, $q_{3}$, and $q_{4}$, corresponding to this experiment, and their desired references $q_{d1}$, $q_{d2}$, $q_{d3}$, and $q_{d4}$, are shown in Figure 20. Note that, based on this figure, the responses of $q_{1}$, $q_{2}$, $q_{3}$, and $q_{4}$ are overdamped. Moreover, for each desired value of $q_{d1}$, the joint $q_{1}$ has a settle time of about 1 s. On the other hand, joints $q_{2}$ and $q_{3}$ have a settle time less than 6 s. It is worth mentioning that the gains of PID controllers for the actuators of joints $q_{2}$ and $q_{3}$ were selected so that their responses are fast enough with the least possible tracking error, despite the gravitational forces acting over them, which are considered as disturbances. Finally, note that $q_{4}$ remains close to its reference $q_{d4}$ = 0°.

Figure 21 shows the velocities ${\hat{\dot{q}}}_{1}$, ${\hat{\dot{q}}}_{2}$, ${\hat{\dot{q}}}_{3}$, and ${\hat{\dot{q}}}_{4}$ estimated by the designed state observers. It can be observed that ${\hat{\dot{q}}}_{1}$, ${\hat{\dot{q}}}_{2}$, ${\hat{\dot{q}}}_{3}$, and ${\hat{\dot{q}}}_{4}$ reach velocities up 41.75, 23.47, 22.5, and 2.3 degrees per second, respectively. These signals are used by the proposed PID controllers of the actuators, whose control signals $u_{1}$, $u_{2}$, $u_{3}$, and $u_{4}$ can be seen in Figure 22. Note that, to reduce the tracking errors, the controllers produce signals $u_{1}$, $u_{2}$, and $u_{3}$ that reach their maximum and minimum values of 24 V and −24 V at some instants of time.

Define the position errors in the Cartesian space as follows:(23)$$\begin{array}{c} \tilde{x}=x_{d}-x\\ \tilde{y}=y_{d}-y\\ \tilde{x}=z_{d}-z\\ \tilde{\phi }=\phi_{d}-\phi \end{array}$$

Similarly, the position error of the *i*th joint is defined as:(24)$${\tilde{q}}_{i}=q_{di}-q_{i},\;i=1,2,3,4$$

Table 5 shows the position error at the instant when the yellow circle is gripped by the robot. Moreover, this table presents the position errors obtained in the remaining three experiments that consisted in collecting a yellow rectangle, a red circle, and a red rectangle. In this table, it is possible to observe that the maximum error in the Cartesian coordinates is less than 6 mm. Table 5 also shows that the maximum error in the joint space is 2.5°.

Table 6 compares the position accuracy of the proposed recycled platform with respect to two platforms based also on the ED-7220C robot. The first platform, described in [45], contains the manufacturer controller of this robot produced by the ED-Corporation company. The second platform, called AUTAREP, uses the controller designed in [12]. A shown in this table, the manufacturer and the AUTAREP platforms have better position accuracy than the proposed recycled platform. We attribute this fact to the mechanical wear of the robot, since it has almost twenty years of service and the backlash in its gears has increased. However, the position accuracy of our proposal is appropriate since the robot gripper opens up to 60 mm, and it collects cylinders with a height of 55 mm and diameter of 40 mm. Despite this accuracy, the proposed platform has fulfilled its aim of allowing undergraduate students to experimentally validate the theory seen in robotics and automatic control courses using the low-cost controller of open architecture. Moreover, the proposed platform has a great utility for educational purposes since their high-level programming based on MATLAB-Simulink permits students to design their own controllers in a simple way and to use several toolboxes to acquire, process, and generate signals for verifying the controller performance. Note also that this software could be used to operate the manipulator as a remote platform without attending to the laboratory, thus allowing its use by students with physical disabilities.

## 11. Conclusions

This article describes a methodology to recycle a 4-DOF educational robot with gripper, which is used for Mechatronics Engineering courses at the University of Colima. Its kinematics, dynamics, and artificial vision are presented along with a proposed low-cost and decentralized controller for the robot joint’s actuators. Furthermore, the process of identifying the parameters of the manipulator actuators is also presented, as well as the use of them for designing PID controllers and implementing state observers that estimate the speed of the actuators. Experimental tests in the manipulator were executed through a proposed graphical user interface that allows selecting the shape and color of the objects gripped by the manipulator. Experiments with the proposed controller were successful in avoiding collisions between the robot and the objects collected by it. It was verified that the maximum positioning errors in the Cartesian and joint coordinates of the robot are 6 mm and 2.5°, respectively. As a future work, we will add another DOF to the manipulator to produce the roll motion in its wrist so that it can pick up objects with an irregular geometry. We will also implement adaptable and robust control schemes in the platform, and we will also upgrade the mechanical components of the manipulator to obtain more precise movements. In addition, we plan to recognize 3D objects with the artificial vision of the robot and to use its GUI for remote practices.

## Figures and Tables

**Figure 1 sensors-20-01694-f001:**
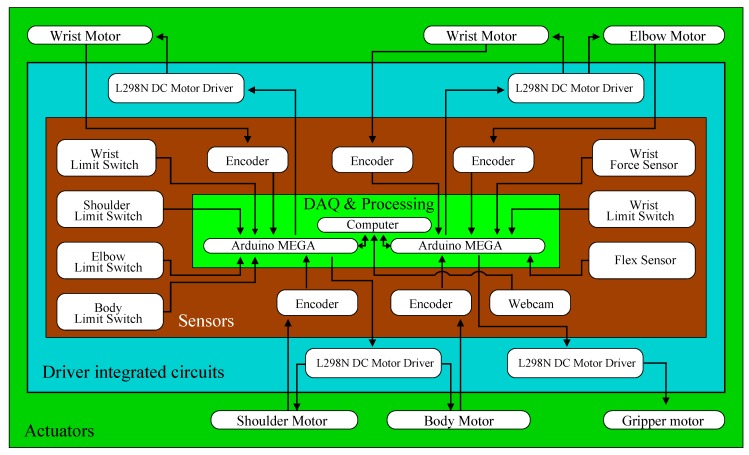
Robot architecture.

**Figure 2 sensors-20-01694-f002:**
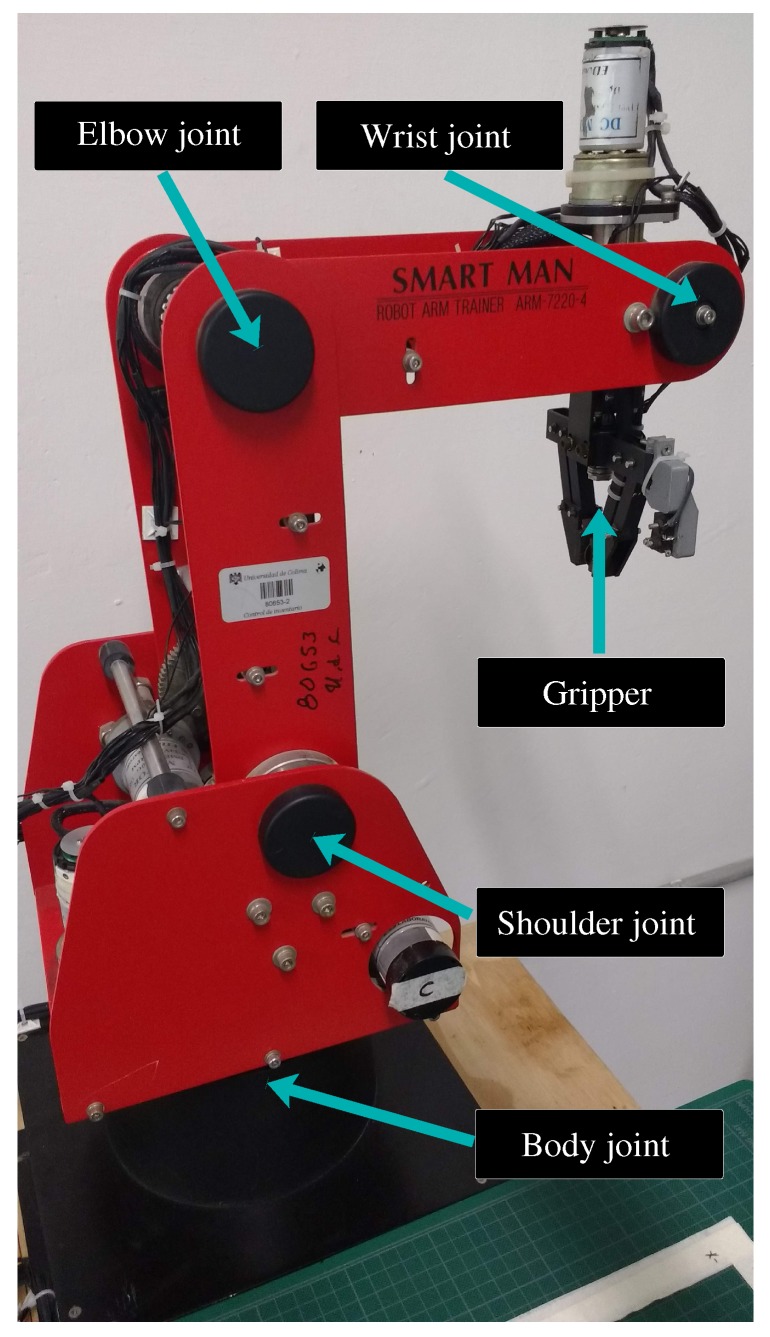
Recycled manipulator.

**Figure 3 sensors-20-01694-f003:**
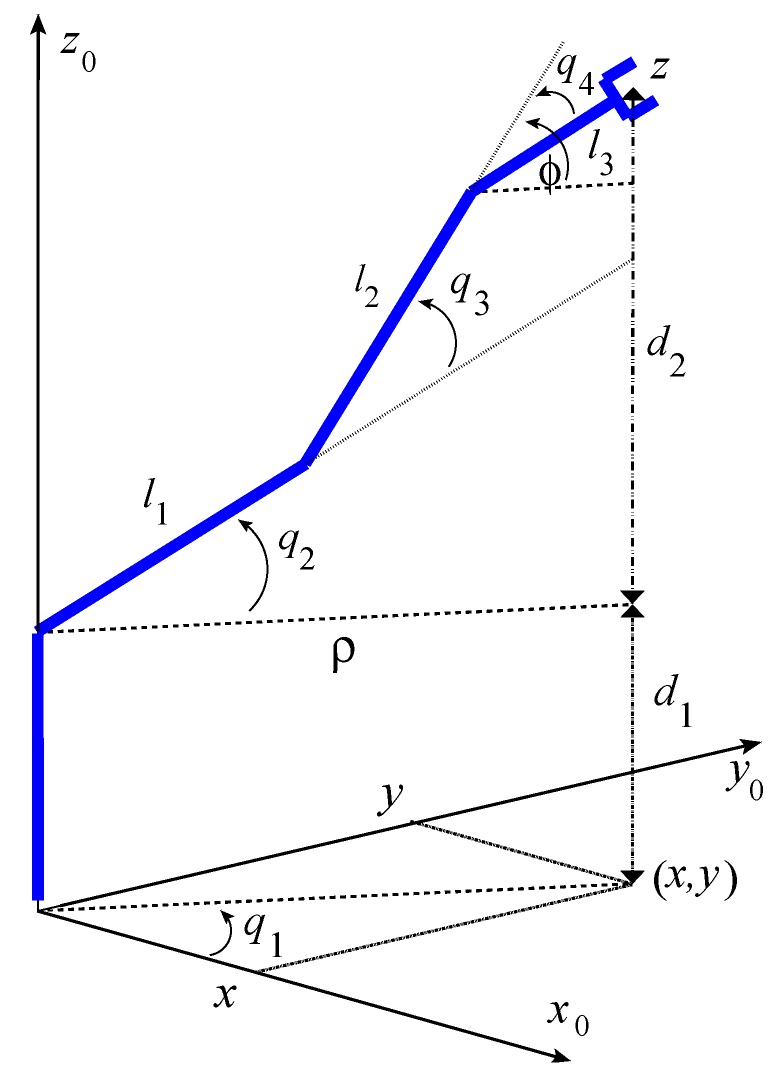
Kinematic scheme of the manipulator.

**Figure 4 sensors-20-01694-f004:**
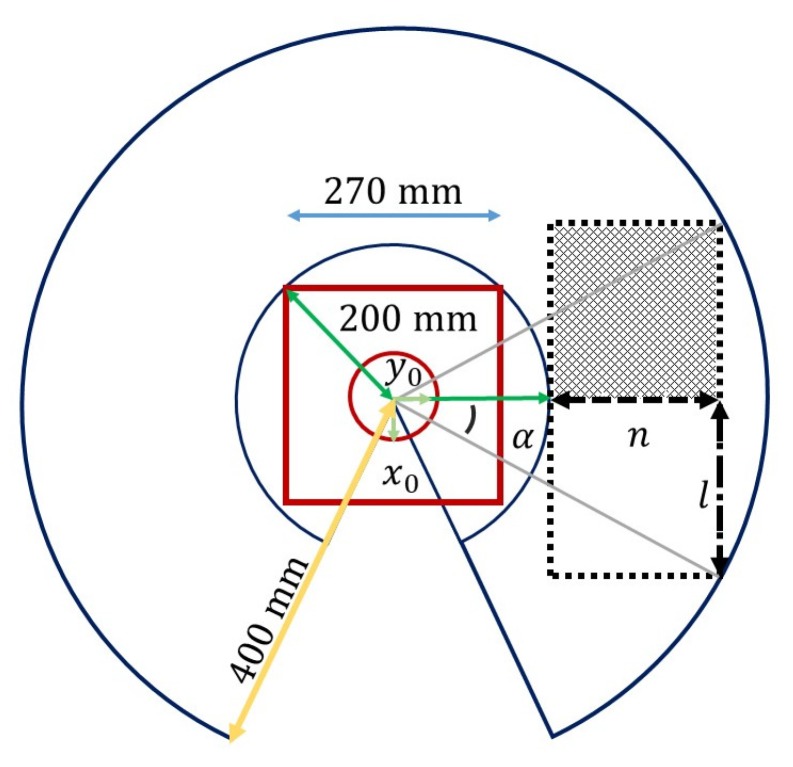
Arm movement range.

**Figure 5 sensors-20-01694-f005:**
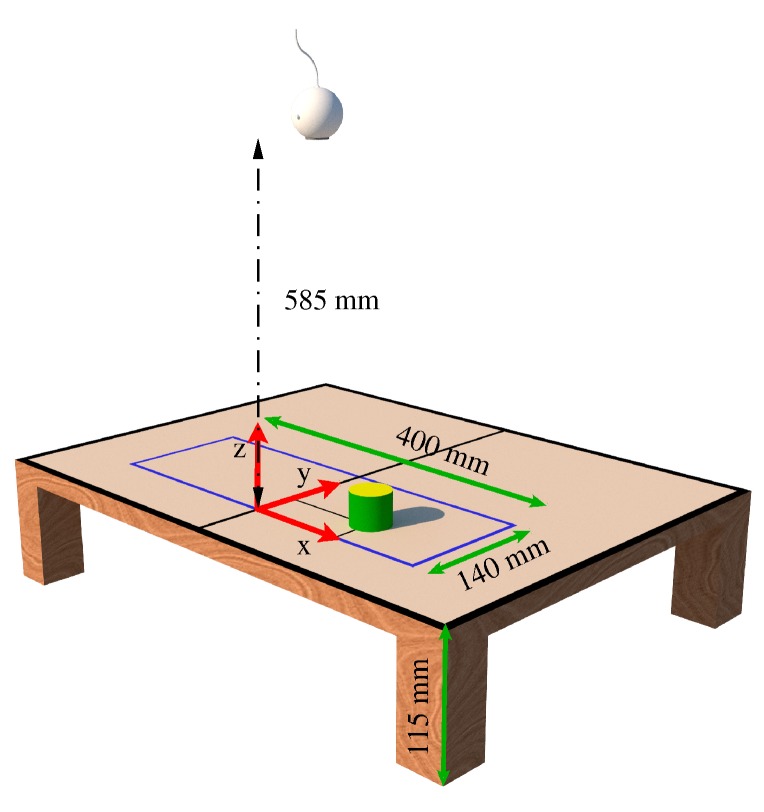
Dimensions of the area where objects are placed.

**Figure 6 sensors-20-01694-f006:**
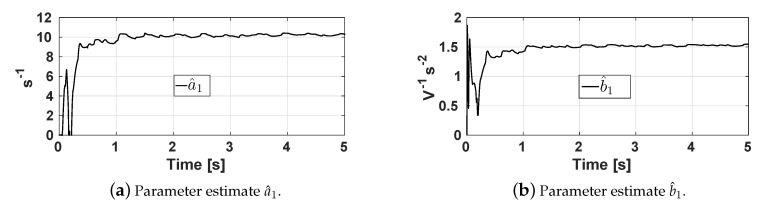
Parameters ${\hat{a}}_{1}$ and ${\hat{b}}_{1}$ estimated by the RLSM.

**Figure 7 sensors-20-01694-f007:**
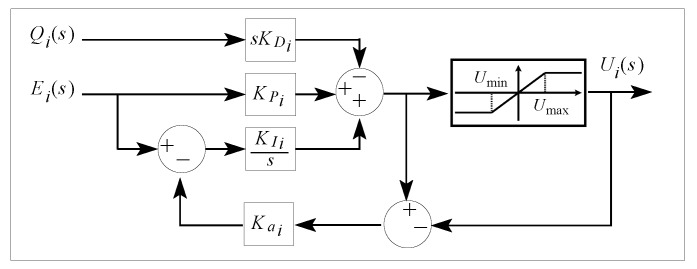
Block diagram of the PID controller with anti-windup compensation.

**Figure 8 sensors-20-01694-f008:**
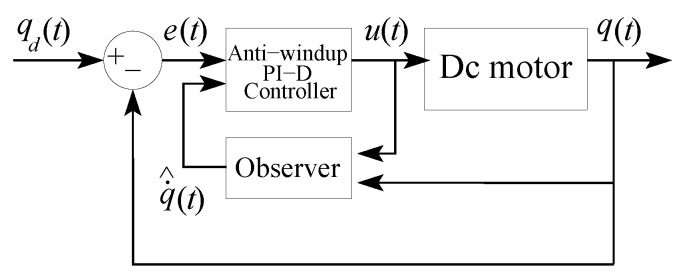
PID controller coupled to the state observer.

**Figure 9 sensors-20-01694-f009:**
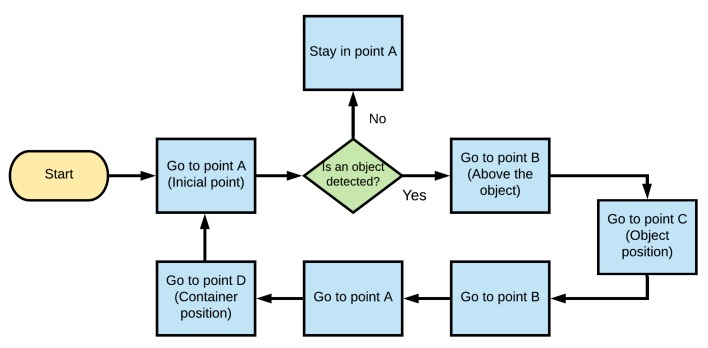
Trajectory planning to move the end-effector.

**Figure 10 sensors-20-01694-f010:**
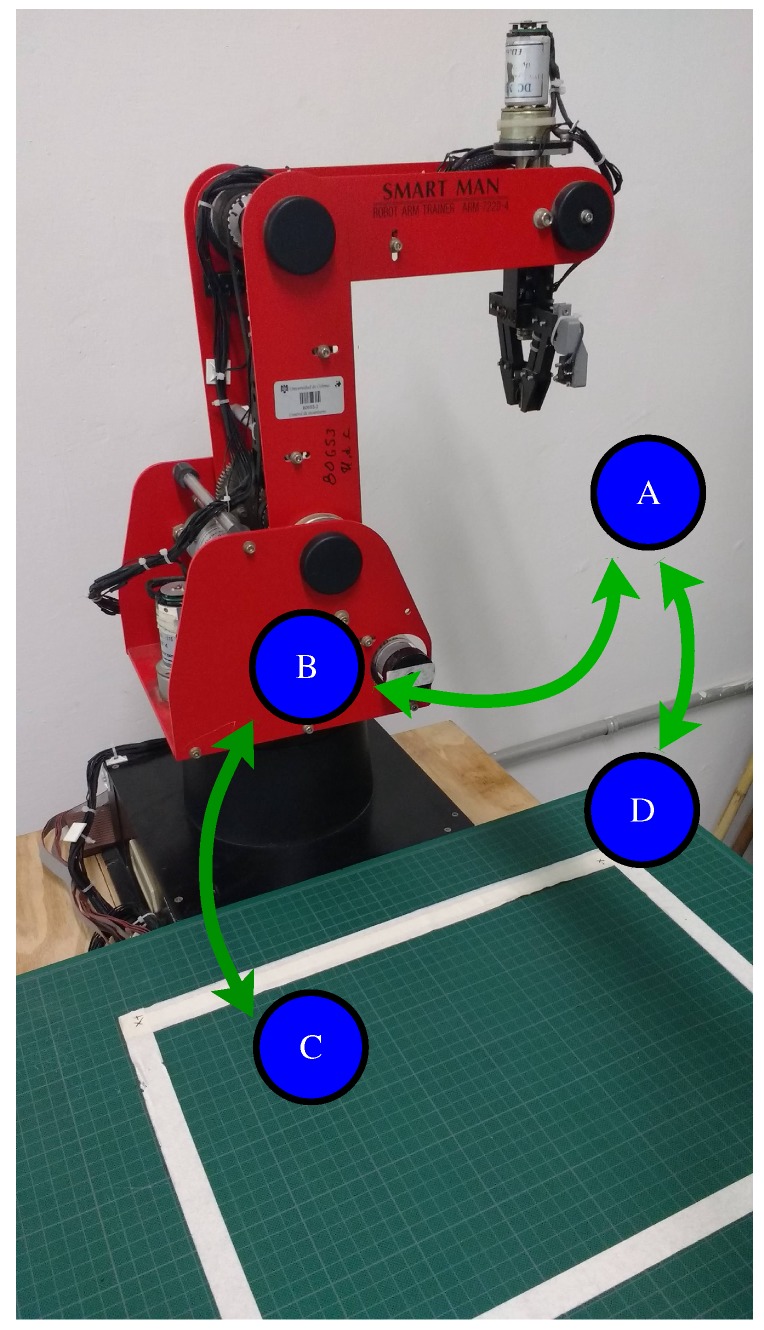
Sequence of Points A–D to move the robot.

**Figure 11 sensors-20-01694-f011:**
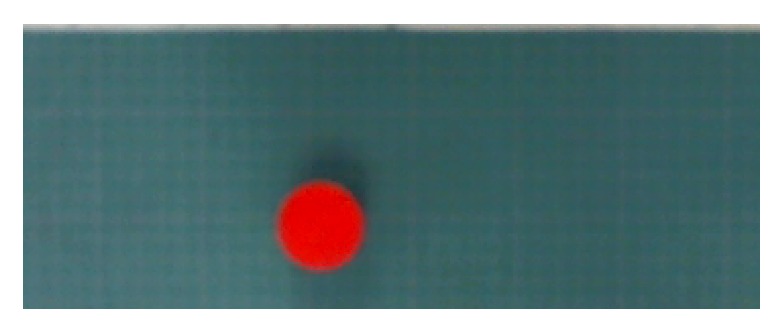
Acquired image of a red circle.

**Figure 12 sensors-20-01694-f012:**
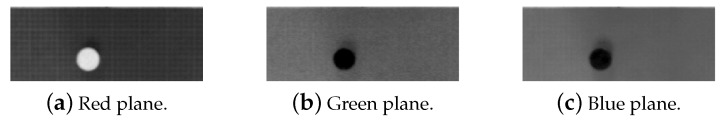
Grayscale of the RGB planes corresponding to the circle in Figure 11.

**Figure 13 sensors-20-01694-f013:**
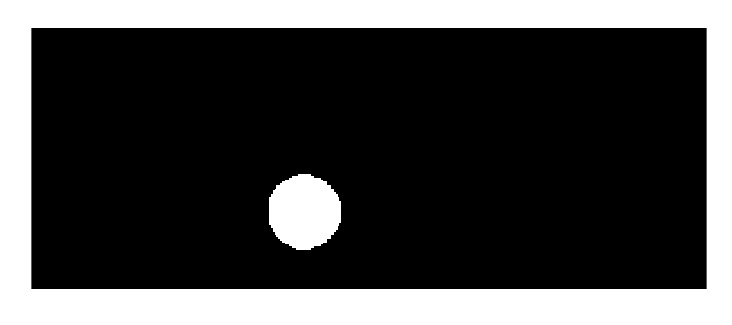
Binary image after the morphological operation.

**Figure 14 sensors-20-01694-f014:**
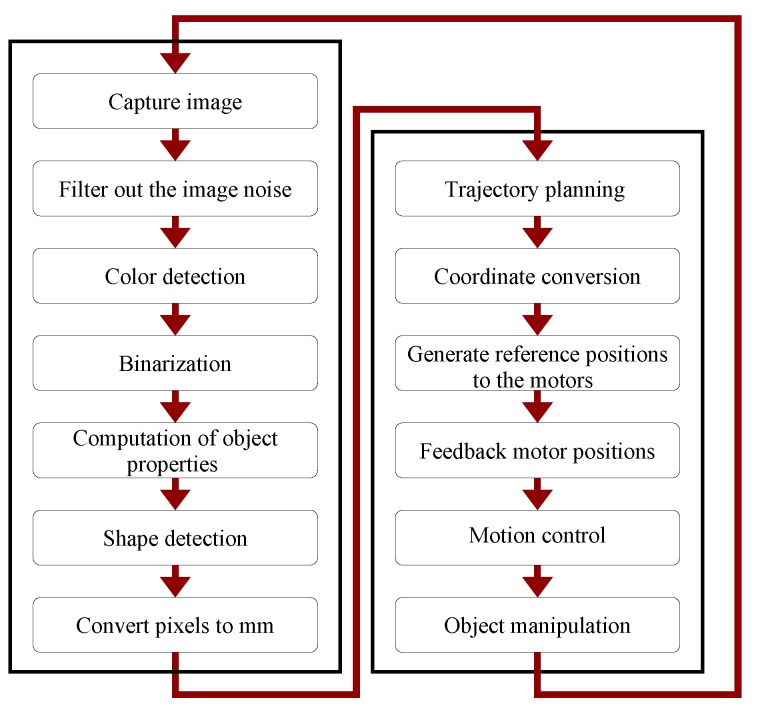
Logic sequence to control the manipulator using artificial vision.

**Figure 15 sensors-20-01694-f015:**
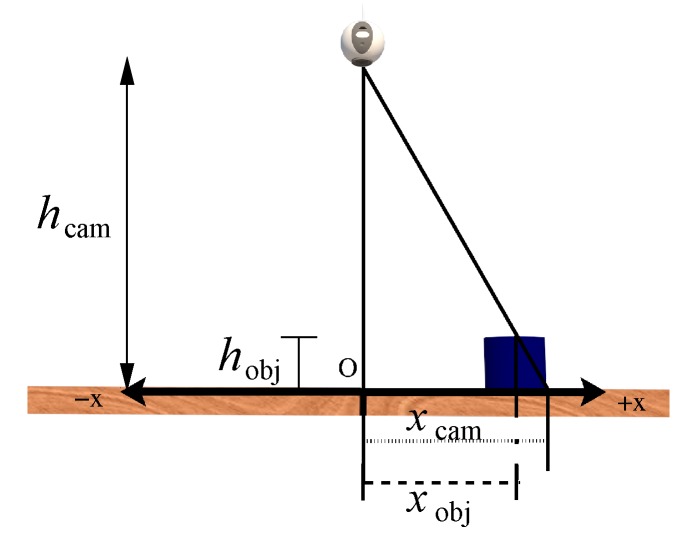
Correction of the object position due to its height.

**Figure 16 sensors-20-01694-f016:**
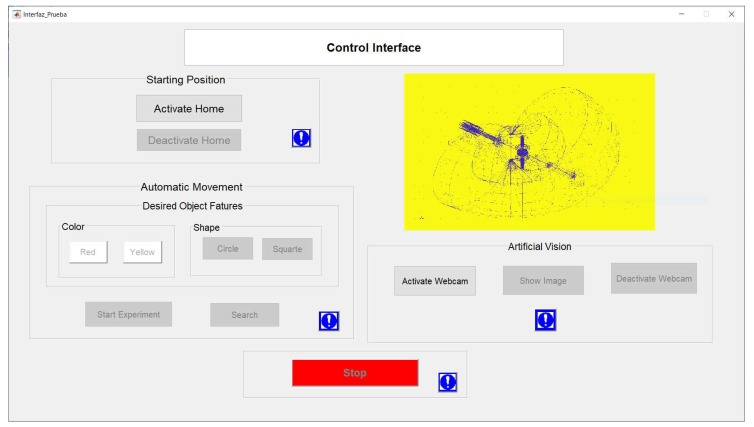
GUI designed to control and visualize the manipulator.

**Figure 17 sensors-20-01694-f017:**
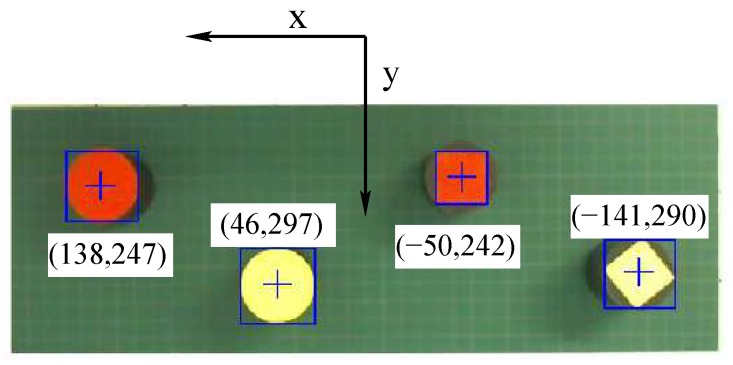
Objects to be collected by the robot.

**Figure 18 sensors-20-01694-f018:**
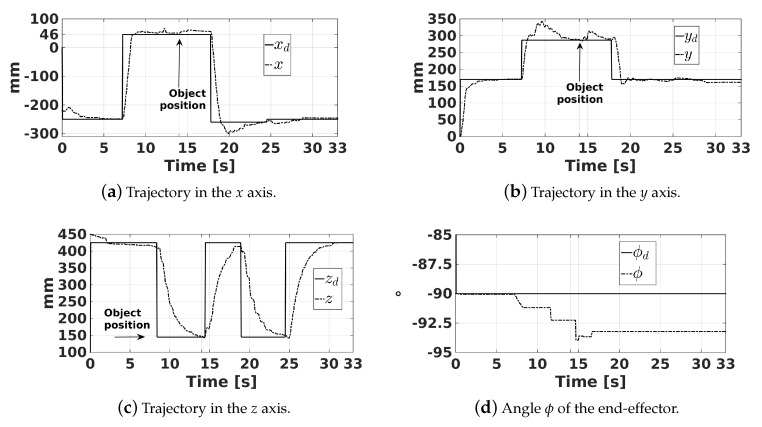
Trajectory of the end-effector.

**Figure 19 sensors-20-01694-f019:**
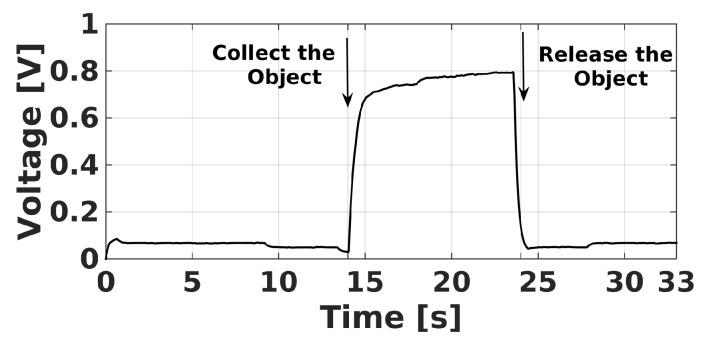
Voltage of the force sensor inside the gripper.

**Figure 20 sensors-20-01694-f020:**
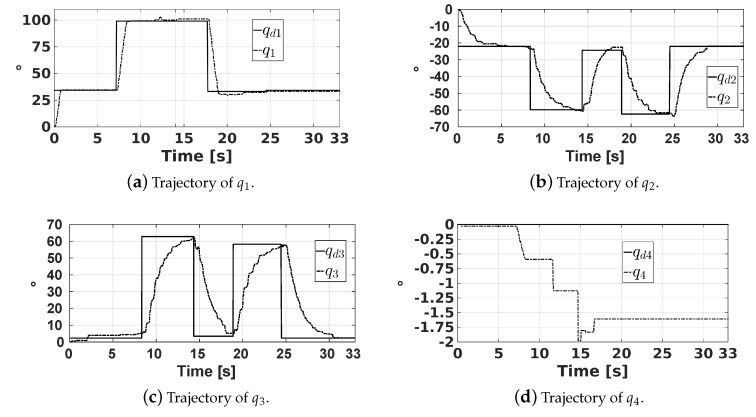
Trajectories of the joint positions.

**Figure 21 sensors-20-01694-f021:**
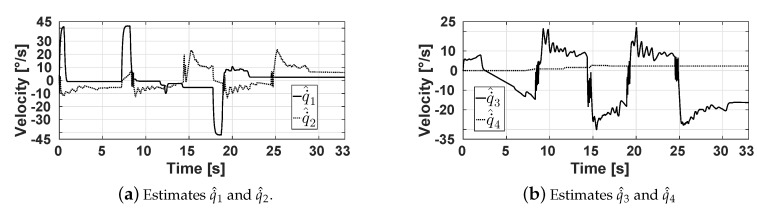
Estimates of the velocity joints provided by the state observers.

**Figure 22 sensors-20-01694-f022:**
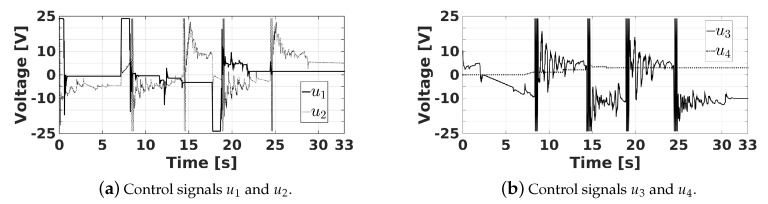
Control signals of the actuators.

**Table 1 sensors-20-01694-t001:** Range for the joints of the manipulator.

Joint	Lower Limit (°)	Upper Limit (°)
$q_{1}$	25	335
$q_{2}$	0	120
$q_{3}$	−66	116
$q_{4}$	−90	−45

**Table 2 sensors-20-01694-t002:** Parameter estimates ${\hat{a}}_{i}$ and ${\hat{b}}_{i}$ of the actuators.

Actuated joint	Estimate ${\hat{a}}_{i}$ [s^−1^]	Estimate ${\hat{b}}_{i}$ [V^−1^s^−2^]
Waist	${\hat{a}}_{1}=10.2$	${\hat{b}}_{1}=1.53$
Shoulder	${\hat{a}}_{2}=13.1$	${\hat{b}}_{2}=1.10$
Elbow	${\hat{a}}_{3}=05.5$	${\hat{b}}_{3}=0.88$
Wrist	${\hat{a}}_{4}=17.0$	${\hat{b}}_{4}=0.88$

**Table 3 sensors-20-01694-t003:** Gains of the PID controllers and state observers.

Actuated Joint	$K_{\mathrm{Pi}}$	$K_{\mathrm{Ii}}$	$K_{\mathrm{Di}}$	$K_{\mathrm{ai}}$	$K_{1i}$	$K_{2i}$
Body	64	0	16	0	13.8	140.65
Shoulder	56	40	20	700	10.9	142.09
Elbow	61	40	18.5	700	18.5	141.41
Wrist	64	0	24	0	7	143.02

**Table 4 sensors-20-01694-t004:** Thresholding values to detect red and yellow colors.

Detected Color	Limits for the Grayscale Image of the Red Plane	Limits for the Grayscale Image of the Green Plane	Limits for the Grayscale Image of the Blue Plane
Red	$p_{\sup\;}^{r}=255$	$p_{\sup\;}^{g}=35$	$p_{\sup\;}^{b}=35$
	$p_{\inf\;}^{r}=200$	$p_{\inf\;}^{g}=0$	$p_{\inf\;}^{b}=0$
Yellow	$p_{\sup\;}^{r}=255$	$p_{\sup\;}^{g}=255$	$p_{\sup\;}^{b}=170$
	$p_{\inf\;}^{r}=200$	$p_{\inf\;}^{g}=200$	$p_{\inf\;}^{b}=0$

**Table 5 sensors-20-01694-t005:** Errors at the instant when the object is taken by the robot.

Object	Cartesian Space Position Errors	Joint Space Position Errors
Yellow circle	$\tilde{x}$ = −5.15 mm	${\tilde{q}}_{1}$ = −1.01°
	$\tilde{y}$ = 0.22 mm	${\tilde{q}}_{2}$ = 0.40°
	$\tilde{z}$ = −1.72 mm	${\tilde{q}}_{3}$ = 1.62°
	$\tilde{\phi }$ = 2.26°	${\tilde{q}}_{4}$ = 1.15°
Yellow square	$\tilde{x}$ = 0.09 mm	${\tilde{q}}_{1}$ = −0.43°
	$\tilde{y}$ = −5.63 mm	${\tilde{q}}_{2}$ = 0.32°
	$\tilde{z}$ = −1.56 mm	${\tilde{q}}_{3}$ = 1.30°
	$\tilde{\phi }$ = −0.17°	${\tilde{q}}_{4}$ = −0.11°
Red circle	$\tilde{x}$ = 0.37 mm	${\tilde{q}}_{1}$ = 0.04°
	$\tilde{y}$ = 0.45 mm	${\tilde{q}}_{2}$ = −0.04°
	$\tilde{z}$ = −2.17 mm	${\tilde{q}}_{3}$ = 1.30°
	$\tilde{\phi }$ = 2.05°	${\tilde{q}}_{4}$ = 1.04°
Red square	$\tilde{x}$ = −1.87 mm	${\tilde{q}}_{1}$ = −0.68°
	$\tilde{y}$ = −5.30 mm	${\tilde{q}}_{2}$ = −0.14°
	$\tilde{z}$ = −2.81 mm	${\tilde{q}}_{3}$ = 2.12°
	$\tilde{\phi }$ = 1.11°	${\tilde{q}}_{4}$ = 0.58°

**Table 6 sensors-20-01694-t006:** Robot accuracy of experimental platforms based on the ED-7220C robot manipulator.

Platform	Accuracy
Proposed recycled robot	6 mm
Manufacturer of the ED-7220C robot	0.5 mm
AUTAREP	2 m

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
