# Peer review of "Recycling and Updating an Educational Robot Manipulator with Open-Hardware-Architecture"

_sensors, 2020, doi:10.3390/s20061694_

Round 1
Reviewer 1 Report
I think most of the concerns have been addressed.
Author Response
We are grateful for your approval.
Reviewer 2 Report
The authors have addressed all comments. It can be accepted.
Author Response
We are grateful for your approval.
Reviewer 3 Report
The previous reviewer remarks have been answered correctly. However, all in all, the contribution is still in a level where its provides a very weak advance to the state of the art in robotics education. There is a lack of a clear explanation of the benefits of the selected approach: for example, some results about the use of the educational robot for a number of students employing it in their studies ?.
Author Response
Point 1:
The previous reviewer remarks have been answered correctly. However, all in all, the contribution is still in a level where its provides a very weak advance to the state of the art in robotics education. There is a lack of a clear explanation of the benefits of the selected approach: for example, some results about the use of the educational robot for a number of students employing it in their studies ?.
Response:
We believe our work will be useful for teams who intend to recycle and upgrade a robot. In our case, we have successful instances, which we described in the new version of the manuscript. Now, at the end of the third paragraph of the Introduction section we added the following text:
The educational robot was also used by three undergraduate students during their final degree projects, whose achievements are reflected in this manuscript. Likewise, the robot has also been used in internal workshops to motivate students to join and remain at the FIME, as well as to show them the importance of robotics and automatic control.
Thanks again, we appreciate your suggestions.
Best regards,
Authors of the manuscript
This manuscript is a resubmission of an earlier submission. The following is a list of the peer review reports and author responses from that submission.
Round 1
Reviewer 1 Report
1)the author should pay attention to recent research. Now except some paper published in Sensors, no recent reference.
Robot Manipulators:
V. Mien, M. Mavrovouniotis and S. S. Ge, "An Adaptive Backstepping Nonsingular Fast Terminal Sliding Mode Control for Robust Fault Tolerant Control of Robot Manipulators IEEE T SYST MAN CY-S 49(7), 1448-1458 (2019). L. Jin, S. Li, L. Xiao, R. Lu and B. Liao, "Cooperative Motion Generation in a Distributed Network of Redundant Robot Manipulators With Noises IEEE T SYST MAN CY-S 48(10), 1715-1724 (2018). C. Yang, Y. Jiang, W. He, J. Na, Z. Li and B. Xu, "Adaptive Parameter Estimation and Control Design for Robot Manipulators With Finite-Time Convergence IEEE T IND ELECTRON 65(10), 8112-8123 (2018). L. Jin, S. Li, X. Luo, Y. Li and B. Qin, "Neural Dynamics for Cooperative Control of Redundant Robot Manipulators IEEE T IND INFORM 14(9), 3812-3821 (2018). D. Chen, Y. Zhang and S. Li, "Tracking Control of Robot Manipulators with Unknown Models: A Jacobian-Matrix-Adaption Method IEEE T IND INFORM 14(7), 3044-3053 (2018). M. Van, S. S. Ge and H. Ren, "Finite Time Fault Tolerant Control for Robot Manipulators Using Time Delay Estimation and Continuous Nonsingular Fast Terminal Sliding Mode Control IEEE T CYBERNETICS 47(7), 1681-1693 (2017). Y. Wang, L. Gu, Y. Xu and X. Cao, "Practical Tracking Control of Robot Manipulators With Continuous Fractional-Order Nonsingular Terminal Sliding Mode IEEE T IND ELECTRON 63(10), 6194-6204 (2016). J. Baek, M. Jin and S. Han, "A New Adaptive Sliding-Mode Control Scheme for Application to Robot Manipulators IEEE T IND ELECTRON 63(6), 3628-3637 (2016).
Artificial vision and vision SLAM
K. Madani, V. Kachurka, C. Sabourin and V. Golovko, "A soft-computing-based approach to artificial visual attention using human eye-fixation paradigm: toward a human-like skill in robot vision SOFT COMPUT 23(7SI), 2369-2389 (2019). K. Madani, V. Kachurka, C. Sabourin, V. Amarger, V. Golovko and L. Rossi, "A human-like visual-attention-based artificial vision system for wildland firefighting assistance APPL INTELL 48(8), 2157-2179 (2018). C. Bousquet-Jette, S. Achiche, D. Beaini, Y. S. L. Cio, C. Leblond-Menard and M. Raison, "Fast scene analysis using vision and artificial intelligence for object prehension by an assistive robot ENG APPL ARTIF INTEL 63, 33-44 (2017). G. Yang, Z. Chen, Y. Li and Z. Su, "Rapid Relocation Method for Mobile Robot Based on Improved ORB-SLAM2 Algorithm REMOTE SENS-BASEL 11(2),(2019). G. Yang, J. Yang, W. Sheng, F. E. F. Junior and S. Li, "Convolutional Neural Network-Based Embarrassing Situation Detection under Camera for Social Robot in Smart Homes. Sensors (Basel, Switzerland) 18(5),(2018).
….
2) The author should compare the platform to other robots, especially in terms of accuracy
3) The author should make it clear why such a platform is needed
4) Experimental results are too weak; The author should present some necessary sound metric, workspace, payload, motion accuracy, motion characteristics, dynamic characteristics… to show the performance of the system.
Reviewer 2 Report
In this paper, the authors proposed a novel approach to recycle and upgrade a 4-DOF educational robot 2 manipulator with a gripper. The topic is quite interesting and this paper is well-written. I have the following comments for the further improvement of the paper.
1) The authors are suggested to add one more keyword, such as robot control, under the abstract.
2) The authors are suggested to describe some recent works on robot control methods and schemes such as the robot motion control via neural networks: New super-twisting zeroing neural-dynamics model for tracking control of parallel robots: A finite-time and robust solution, IEEE Transactions on Cybernetics, DOI: 10.1109/TCYB.2019.2930662, and the service robot control: Multi-DOF counterbalance mechanism for a service robot arm, IEEE/ASME Trans. Mechatronics, vol. 19, no. 6, pp. 1756–1763, Dec. 2014.
3) Some important figures are unclear. The authors are suggested to present Fig. 2 and Fig. 9 in larger size.
4) The authors are suggested to upload the codes and demos to the open-source website such as GitHub.
The paper is good writing and presents technical contributions, which could be accepted after a minor revision.
Reviewer 3 Report
This contribution is aiming to provide an educational robot. However, being this subject very relevant, the contribution has several weak points:
1) There are not given references of previous author's work in this area. However there is a recent publication:
"CONCHA-SANCHEZ, Antonio, FIGUEROA-RODRÍGUEZ, Juan Felipe, FUENTESCOVARRUBIAS, Andrés Gerardo y FUENTES-COVARRUBIAS, Ricardo: Low cost experimental platform for decoupled control from a 5 DoF manipulator robot. Revista de Tecnologías en Procesos Industriales. Septiembre 2018 Vol.2 No.4 1-11". ISSN: 2523-6822.
This publication, although in different language, has very similar content than the pages 1 to 6 of this contribution to Sensors. So, having provided no references, or just a summary of previous work, the contribution cannot be considered, in this part, "original".
2) The contribution provides a series of steps toward the educational, recycled, robot. However it lacks the necesary "glue" to connect all of them. For example, there is a big jump from section 6 on robot control, to 7 on trajectory planning: there is not an explanation of why to do that.
3) In the abstract and in other sections, there is a mention of a 4-DOF robot. However, model ED-7220C (ARM TRAINER ROBOT), is a 5-DOF + gripper. This needs a clarification?.
Moreover, in Figure 1, there are mentioned 6 motors (I assume, 5 for robot (3 for positioning and 2 for the pitch and roll of the hand, + the open/closing gripper motor). Please revise.
4) There is one mistake in equation (11), there is a "-" for the "D" term of the PID. It should be "+".
5) The section for artificial vision (sect. 8) does not provide a clear understanding of the robot posibilities, because using MATLAB/SIMULINK it opens a very wide area for the educational use. The same for section 9 and for section 10.
In this sense the contribution fails to achieve its goals, it shows and arrangement of the robot itself, but the usability of the system for educational purposes is not offered.